# Essential Oil Headspace Volatiles Prevent Invasive Box Tree Moth (*Cydalima perspectalis*) Oviposition—Insights from Electrophysiology and Behaviour

**DOI:** 10.3390/insects11080465

**Published:** 2020-07-23

**Authors:** Magdolna Olívia Szelényi, Anna Laura Erdei, Júlia Katalin Jósvai, Dalma Radványi, Bence Sümegi, Gábor Vétek, Béla Péter Molnár, Zsolt Kárpáti

**Affiliations:** 1Zoology Department, Plant Protection Institute, Centre for Agricultural Research, 1022 Budapest, Hungary; szelenyi.magdolna@agrar.mta.hu (M.O.S.); erdei.anna.laura@agrar.mta.hu (A.L.E.); radvanyi.dalma@agrar.mta.hu (D.R.); karpati.zsolt@agrar.mta.hu (Z.K.); 2Department of Applied Chemical Ecology, Plant Protection Institute, Centre for Agricultural Research, 1022 Budapest, Hungary; josvai.julia@agrar.mta.hu; 3Department of Applied Chemistry, Faculty of Food Science, Szent István University, 1118 Budapest, Hungary; 4Department of Entomology, Faculty of Horticultural, Szent István University, 1118 Budapest, Hungary; ben-c_05@hotmail.com (B.S.); Vetek.Gabor@kertk.szie.hu (G.V.)

**Keywords:** essential oil, volatile profile, lavender, cinnamon, eucalyptus, oviposition-deterrents, invasive species, *Cydalima perspectalis*, Lepidoptera, electrophysiology

## Abstract

The box tree moth (*Cydalima perspectalis* Walker) is an invasive species in Europe causing severe damage both in natural and ornamental boxwood (*Buxus* spp.) vegetation. Pest management tactics are often based on the use of chemical insecticides, whereas environmentally-friendly control solutions are not available against this insect. The application of essential oils may provide effective protection against oviposition and subsequent larval damage. Oviposition deterrence of cinnamon, eucalyptus and lavender essential oils was tested on female *C. perspectalis* in behavioural bioassays. Our results indicate that all the studied essential oils may be adequate deterrents; however, cinnamon oil exhibited the strongest effect. To determine the physiologically active compounds in the headspace of the essential oils, gas chromatography coupled with electroantennography recordings were performed in parallel with gas chromatography-mass spectrometry to identify the volatile constituents. In addition, the release rates of various components from vial-wick dispensers were measured during the oviposition bioassay. These results may serve as a basis for the development of a practical and insecticide-free plant protection method against this invasive moth species.

## 1. Introduction

Invasive pests represent a great concern for the native ecosystem on their newly conquered territories. Global trading facilitates the spread of non-native, invasive alien species at an alarming rate [1,2]. The box tree moth (BTM) (*Cydalima perspectalis* Walker, Lepidoptera: Crambidae) has recently been introduced into Europe and has rapidly become a highly invasive and destructive pest, feeding exclusively on boxwood (*Buxus* spp.) and different varieties [3,4,5,6]. Females lay their eggs on boxwood leaves, and larval feeding may lead to complete defoliation of the shrubs. Besides being a pest of major concern in urban environments all over Europe, in the western, southern and eastern parts of the continent, large areas of natural boxwood forests are also at a severe risk posed by this pest [7,8,9]. Because of its non-native origin, BTM has only a limited number of natural enemies in Europe [6]. Chemical plant protection is extremely challenging due to the mobility of adults and the hidden lifestyle of the larvae [6]. The sex-pheromone of the species has already been identified and has proved to be a useful tool for monitoring males [10,11]. However, due to the extremely high population density of BTM on the invaded territories even the so-called non saturable, high-capacity funnel traps overflow very quickly, and therefore the maintenance of traps requires intensive manual labour [12].

Repellents act locally or at a distance, preventing arthropods from flying toward, landing upon, or feeding on the host [13]. Usually, insect repellents work by providing a vapour barrier around the host, deterring the arthropod from coming into contact with the plant surface [14]. Therefore, deterring egg-laying by BTM females by the application of natural repellents would be an environmentally sound pest control tactic. To date, three volatile components of the BTM larval frass have been identified as potential oviposition-deterrents. Although the synthetic blend of these three compounds significantly reduced the number of laid eggs, it did not completely prevent the oviposition and the damage [15].

For potential sources of repellents, thousands of plants have been screened in the last 50 years [16], and recently essential oils (EOs) became the focus of attention. These concentrated hydrophobic liquids can be isolated from a vast array of plants, and the composition highly depends on the genotype [17], on the phenological stage [18], on the harvested plant part [19] and on the extraction method [20]. The volatile components are biosynthesized in different pathways, so they belong to polyketides and lipids, shikimic acid derivatives and terpenoids (hemi-, mono, sesquiterpenoids) [16,17]. These complex mixtures may contain over 300 different compounds [21], generally with low molecular weight [22]. In spite of their complexity, most of the EOs are dominated by two or three major components while the rest of the compounds are present only in trace amounts [17].

The antifeedant effect of EOs was studied against lepidopteran (*Trichoplusia ni* Hübn. [23]) and coleopteran (*Leptinotarsa decemlineata* Say) [24] species, however, contact application on plants is limited due to the phytotoxic properties of EOs. In fumigation, the volatile compounds are responsible for lethality. The fumigant efficiency of different EOs was investigated against storage pests, e.g., towards the rice weevil *Sitophilus oryzae* L. [25] and *Plutella xylostella* L. [26]. To investigate the contact application of EOs regarding their oviposition deterrence, research was concluded e.g., on medfly (*Ceratitis capitata* Wiedemann) [27] and on cowpea beetle (*Callosobruchus maculatus* F.) [28]. Studies were also conducted on lepidopteran species such as *Phthorimaea operculella* Zell. [29], *Anticarsia gemmatalis* Hübn. [30], *Spodoptera littoralis* Boisd. [31], *Spodoptera frugiperda* Smith [32], and *Tuta absoluta* Meyrick [33], showing that EOs have the potential to modify the egg-laying behaviour both as contact and as volatile deterrents. Concerning the ovipositing behaviour of BTM, only one study investigated the contact ovipositing deterrent effect of different EOs [34].

Some plant-based repellents are comparable to, or even more effective than, synthetic volatile blends, but EOs can be phytotoxic and, in dispensers, they tend to be short-lived in their efficacy. Consequently, the development of dispensers for long-lasting EO emission both indoors and outdoors has become more and more important both in the food industry [35] and plant protection, like encapsulated lemongrass EO [36], or the nanoemulsion of eucalyptus oil [37].

Our first aim was to assess the repellence of three essential oils selected from different plant families: cinnamon (*Cinnamomum verum* Schaeff., Lauraceae), eucalyptus (*Eucalyptus globulus* Labill., Myrtaceae), and lavender (*Lavandula angustifolia* Mill., Lamiaceae) on BTM females by performing oviposition bioassays in laboratory conditions. Secondly, using gas-chromatography-coupled electroantennographic detection (GC-EAD), we searched for components of the EO headspace that could result in electrophysiological responses by the female BTM antennae. Third, we identified the headspace volatile components of the three EOs with gas-chromatography-coupled mass spectrometry (GC-MS). Finally, we measured the temporal emission dynamics of dispensers filled with EOs using solvent-free solid phase microextraction (SPME).

## 2. Materials and Methods

### 2.1. Insects

Box tree moths were collected in an early larval stage from public gardens at different locations of Budapest, Hungary. The larvae were kept in a climate chamber (25 ± 1 °C, 65 ± 5% RH, with a photoperiod of 16L:8D) to establish a laboratory colony. Larvae were fed on 5–7 boxwood shoots placed in cylindrical glass jars (I.D. 20 × 100 cm) equipped with small dishes containing water. For the experiments, the pupae were collected and sexed. For experimenting, male and female pupae were placed in separate rearing cages covered with mesh until emergence. For rearing, male and female pupae were left to emerge together and potted boxwood plants were offered for egg-laying. Leaves with eggs were replaced into cylindrical glass jars, as described above, where boxwood shoots were offered for the hatching larvae to feed upon and develop.

### 2.2. Essential Oils and Dispensers

The following commercially available high-purity essential oils were tested in the experiments: lavender (100% purity); cinnamon (100% purity); and eucalyptus oil (100% purity) (Naturol, Fotima-Plus Ltd., Budapest, Hungary).

Because of the high volatility, potential cytotoxicity and/or phytotoxicity (herbicide) of EO components on plant cells [13,18,36], EOs were not sprayed directly onto the foliage of boxwood. Instead, vial-wick dispensers were hung onto the shoots and used during the oviposition bioassays. Vial- wick dispensers were made using amber glass vials (2 mL) with open-top caps. Through a hole in the PTFE/silicone septa, the top of a PTFE tube (I.D. 2 mm, length 40 mm) with a cotton wick (total length 50 mm) was exposed to the air by 4 mm, while the other end was dipped into the essential oil (1.5 mL) inside the vial [18]. This ensured a long-lasting volatile release whilst avoiding plant damage. The same type of dispenser was used for longevity analysis.

### 2.3. Oviposition Bioassay

To determine the behavioural effect of EOs on female oviposition, the bioassays were conducted in parallel in two screened cages with polyester netting (100 × 100 × 100 cm) (Figure 1) in a climate room (25 °C, 60% RH, 16L:8D). One day prior to the experiment, two boxwood plants (*Buxus sempervirens* L. ‘Suffruticosa’) were placed into the cage. The plants were of the same variety, age, size and phenological stage, and potted in the same type of two-liter nursery container. The offered boxwood variety has dense foliage and a naturally spherical shape; the average height and width of the plants were 32 and 28 cm, respectively. Each plant was equipped with three vial-wick dispensers. Vials were placed at the bottom, middle and upper part of the plants (as shown in Figure 1). After this procedure, 5-5 freshly emerged male and female moths were released in the cages. The position of the plants within the cages was rotated randomly once a day.

The EO-treated plants were tested in pairs with untreated control plants (lavender vs. control, cinnamon vs. control, and eucalyptus vs. control). Leaves with eggs were collected from the plants and the oviposition response was determined by counting the numbers of eggs laid on leaves in five consecutive days. The trials were repeated four times except in the case of lavender, where only three trials were conducted.

### 2.4. SPME Volatile Collections and GC-MS Analysis

Volatile headspace collections were conducted from the three above-mentioned commercially available undiluted essential oils, as were tested in oviposition bioassay. A total of 5 µL of the essential oil was pipetted into 4 mL vials and closed with a screw cap equipped with PTFE/silicone septa. After 10 min, the SPME fiber (DVB/CAR/PDMS 50/30 µm, Supelco, Sigma-Aldrich, Bellefonte, PA, USA) was pierced through the septum and exposed to the headspace of the sampling vial for 5 min at room temperature (20 ± 1 °C) in three replicates. The SPME fiber was placed into the GC-MS (HP Agilent 6890 GC and 5975 MS, Agilent Technologies, Palo Alto, USA) injector port for thermal desorption. The injector temperature was set to 250 °C in splitless mode and the split valve was closed for 1 min before septum purge. The GC was equipped with HP-5 UI capillary column (30 m × 0.25 mm × 0.25 µm, J&W Scientific, Folsom, CA, USA). Helium was used as the carrier at 35 cm s^−1^ flow. The oven temperature was initially held at 50 °C for 1 min, then raised to 260 °C at 10 °C min^−1^, where it was held for 10 min. The transfer line temperature was 280 °C. The mass spectrometer was operated in electron impact (EI) ionization mode at 70 eV, scanning m/z 29–400, at 2 scans s^−1^. Before each measurement, fibers were conditioned at 250 °C in the split/splitless injector of the GC-MS in split mode for 10 min.

Compounds were tentatively identified by matching their mass spectra with those in the MS Libraries (NIST 11 and Wiley) and the identification was verified by the comparison of calculated Kováts index (Ki) values with those published in NIST Chemistry WebBook database. For performing statistical analysis and calculating relative abundances, manually integrated areas from the total ion chromatogram adapted to the baseline were used.

### 2.5. Electrophysiological Experiments of SPME Headspace (GC-EAD)

Coupled gas chromatographic–electroantennographic detection (GC-EAD) was performed on female box tree moths following the procedure described by Molnár et al. [38]. Briefly, two-day-old mated female antennae were used to pinpoint the antennal active volatile components of the three essential oils. The GC was equipped with an HP-5 capillary column (30 m × 0.32 mm × 0.25 µm, J&W Scientific) and was used with a split/splitless injector. The injector was used in splitless mode for 0.5 min at a temperature of 270 °C. The oven temperature was held at 50 °C for 1 min and then increased by 10 °C min^−1^ up to 230 °C. Helium was used as the carrier at a constant flow rate of 2.9 mL min^−1^. The GC effluent was equally split to the FID (280 °C) and to the heated EAD port (220 °C). The EAD effluent was delivered into a stream of charcoal-filtered and humidified air (1 L min^−1^) and led to the antennal preparation.

Headspace SPME volatile collection was injected into the GC. The antennal signal was pre-amplified 10 times, converted to a digital signal by a high-input impedance DC amplifier interface (IDAC-2, Syntech) and recorded simultaneously with the FID signal on a computer using a GC-EAD 2012 software (version 1.2.4, Syntech). Experiments were designed with five replicates of each essential oil.

### 2.6. Dispenser Release Rate and Longevity Analysis by SPME Sampling

The release rate of the dispensers loaded with EOs was measured on a daily basis for five consecutive days to mirror the duration of the oviposition bioassay. Three independent dispensers were measured for each EO. The headspace of EOs was sampled with SPME fibers, as described above. Prior to sampling, the vial-wick dispensers filled with 1.5 mL of EO were placed in glass vials (L: 80 mm, ID 20 mm), and 5 min before sampling the vials were sealed with aluminium foil and laboratory film (American Can, Greenwich, CT, USA). The SPME fibers were exposed to the closed headspace of the sampling vial for 5 min at room temperature and subsequently analyzed by GC-MS. The abundance of antennal active volatile compounds shared among tested EOs was monitored: 1-terpinen-4-ol, *β*-pinene, *cis*-linalool oxide, (±)-linalool and terpinolene. Before each measurement, fibers were conditioned at 250 °C in the split/splitless injector of the GC-MS (HP Agilent 5890 GC and 5975 MS, Agilent Technologies, Palo Alto, USA) in split mode for 10 min.

### 2.7. Statistical Analysis

In the case of the oviposition bioassay, one-way ANCOVA was used to investigate the effect of treatment (as factor) and time (as covariant) on the mean number of eggs laid on plant leaves. Assumptions of ANCOVA were checked by plot diagnosis.

Oviposition indices (Oi) for the three EOs were calculated as (O−C)/(O+C), where O was the number of eggs laid on treated plants, and C is the number of eggs laid on control plants [39]. Oviposition indices were compared using one-way ANOVA. As homogeneity of variances was not assumed (Levene’s test, *p* > 0.05), oviposition indices were separated using the Games–Howell post hoc test. Statistical analyses were performed in R 3.6.3 [40].

Univariate GLM was performed to describe the change in the release rate for the chosen compounds (as a dependent variable) during the time of the experiment (as covariant). Preliminary analyses were performed to ensure there was no violation of the assumption of normality and linearity. Statistical procedures were conducted by using IBM SPSS Statistics for Windows, Version 22.0 (Armonk, NY, IBM Corp.).

## 3. Results

### 3.1. Oviposition Bioassay

Each tested EO significantly reduced the number of eggs laid upon treated as compared to control plants (cinnamon: F_1,36_ = 26.39, *p* < 0.001; eucalyptus: F_1,36_ = 16.71, *p* < 0.001; lavender: F_1,26_ = 7.64, *p* < 0.05). Time had no effect on the repellency; EOs performed with equivalent efficiency throughout the five days of the trials.

The calculated oviposition indices of EOs were significantly different (Figure 2). The highest deterring effect was observed in the case of cinnamon; about 75% fewer eggs were found on the treated plant as compared to the control. Eucalyptus oil also reduced the number of eggs laid upon treated plants, though to a lesser degree, and lavender caused the lowest effect among the three EOs.

### 3.2. Chemical and Chemosensory Characterisation of EOs

#### 3.2.1. Cinnamon

By using SPME headspace sampling and GC-MS, 67 chemical compounds were separated and identified from the cinnamon oil comprising 98.39% of the total volume. The dominant constituents were *trans*-cinnamaldehyde (49.82%), *p*-cymene (10.52%), benzaldehyde (8.23%), *α*-phellandrene (7.56%), benzenepropanal (2.07%), *α*-copaene (2.01%), 2-anisaldehyde (1.88%), *α*-pinene (1.64%), *cis*-ß-ocimene (1.50%), styrene (1.23%), phenylethyl alcohol (1.07%), cinnamyl acetate (1.04%) (Appendix A).

Ten volatile compounds of the cinnamon oil headspace evoked constant and robust responses from female BTM antennae (Figure 3 and Table 1).

#### 3.2.2. Eucalyptus

In eucalyptus oil, eucalyptol (76.96%), an oxygenated monoterpene, was the most abundant compound, and only *α*-pinene (8.31%), β-pinene (1.54%), *α*-terpineol (1.47%), *cis*-limonene-oxide (1.28%), D-limonene (1.09%), *p*-cymene (1.02%) and (±)-linalool (0.05%) were present with more than 1% relative abundance. Others like terpinolene, isoamyl valerianate, *α*-campholenal, *trans*-limonene-oxide, *trans*-carveol and carvone also occurred in higher than 0.5% relative abundance (Appendix A). The 54 identified components represent 99.73% of the total eucalyptus oil blend.

Eight volatile compounds of the eucalyptus oil headspace evoked constant responses from female BTM antennae even though most were present only with low abundances (Figure 4 and Table 1).

#### 3.2.3. Lavender

A total of 55 compounds were identified in the lavender essential oil representing 96.15% of the total oil. The main constituents were: (±)-linalool (24.07%), eucalyptol (12.62%), linalyl acetate (11.76%), camphor (11.70%), *β*-pinene (6.70%) and *α*-pinene (4.92%). Others like acetic acid, D-limonene, borneol, neryl acetate and β-caryophyllene also occurred in higher than 1.5% relative abundance (Appendix A).

In GC-EAD recordings, 13 volatile compounds of the lavender oil headspace evoked reproducible responses from female BTM antennae (Figure 5 and Table 1).

### 3.3. Temporal Changes of the Dispenser Emitted Volatiles

As the coefficients of the linear model indicate, the release of all the five compounds increased significantly in cinnamon oil. In eucalyptus and lavender oil, all of the compounds decreased significantly, except *β*-pinene in eucalyptus and terpinolene in lavender (Appendix A, and Table 2).

## 4. Discussion

Herbal essential oils have largely been considered as repellents for a vast array of arthropod species. EOs contain several volatile organic compounds from diverse chemical groups, which could play an important role in the repellent effect observed by several authors [41].

In the present study, BTM females laid the fewest eggs on the box tree treated with cinnamon oil, i.e., the cinnamon oil has the highest deterrent effect. In the case of cinnamon oil, the repellent effect has been shown for mites [42], mosquitoes [43] and codling moth larvae [44]. Fouad et al. [45] compared five EOs, and the cinnamon oil was the most repellent oil against cowpea beetle (*Callosobruchus maculatus F.*). By using electroantennography (EAG) on the African malaria mosquito (*Anopheles gambiae Giles*), the antennae gave the highest response to synthetic cinnamaldehyde, which is the most abundant component in cinnamon oil [46]. In our case, trans-cinnamaldehyde was also the most abundant component in cinnamon oil; however, the BTM antennae did not give a distinguishable response to this compound (Figure 3), therefore, this component is not necessarily responsible for the deterrent effect shown. In cinnamon oil, the most intensive physiological response was given for benzaldehyde, although it was not the most abundant component in this oil. Ayyub et al. [47] showed that benzaldehyde, among other aldehydes, had the strongest repellent effect for *Drosophila melanogaster* (Meigen).

In the eucalyptus oil, GC-MS measurements show that the most abundant peak, eucalyptol, failed to elicit an antennal response from the BTM female antennae, indicating that the most abundant components are not essential to induce the oviposition deterrent effect (Figure 3 and Figure 4). Eucalyptus oil is widely studied insect repellent [48], and it also showed an oviposition-deterrent effect against *Plutella xylostella* (L.) [49]. Although eucalyptol is a characteristic compound in eucalyptus oils and is used as a repellent against many insects [50], there was no detectable response towards it from BTM.

Although the highest amount of electrophysiologically active compounds were found in lavender, it proved to be the least effective in deterring egg-laying behaviour. Lavender oil has been tested as a potential oviposition deterrent agent against BTM [34], where lavender did not show any effect on ovipositing behaviour. The different outcomes of our study could be explained with the different concentrations and dispersion methods used. In both studies, linalool was the most abundant component. In our study, the highest EAD response was also evoked by linalool; moreover, (±)-linalool occurred in all three EOs tested in this study, and gave high EAD responses in every measurement. It would be logical to infer that this component, alone or in combination with other antennal active components, may be the compound underlying the observed behavioural effect (Figure 5). Molnár et al. [15] showed that a synthetic blend of linalool, guaiacol and veratrol identified from BTM larval frass could also deter oviposition. To account for this discrepancy between the lower efficiency of lavender oil and these previous findings, it is important to consider that this compound is a common component of floral scents and we used a lavender oil extracted from the flowers of the plant. Odours of the lavender flower can also act as feeding attractants for the female BTM [51,52]; however, the flower emits these compounds in a lower concentration compared to the commercial EOs, and deterrence just as attraction can be dose-dependent. Therefore, the high concentration of linalool could still be a deterrent for females. We have to consider that linalool can evoke different behavioural effects in the case of different species: for instance, in the case of sphinx moth (*Manduca sexta* L.) linalool did not affect the feeding behaviour; however, the females laid fewer eggs on plant emitting one of the enantiomers of linalool [53].

The results of electrophysiological and behavioural bioassays are strongly dose-dependent [54], hence we cannot rule out the possibility that other behaviorally active compounds were also present in the EOs, but their abundance was not high enough to elicit distinguishable antennal response. To fully understand the underlying reasons for the observed deterrent effect, we also have to consider the qualitative differences between the antennal active compounds in the three EOs. Our results also highlight the phenomenon that the most abundant components are not always responsible for the behaviour, which has been demonstrated also on *Anopheles gambiae* (Giles) [55]. The long-term stability of EOs—especially if they are exposed to direct sunlight or other environmental factors—is not sufficient for their direct application, as the ratio of behaviourally active components could alter quickly, resulting in the loss of behavioural effect [55]. Dispensers containing the key compounds in the optimal dose and composition would be a more reliable alternative for the application in pest management. Therefore, we compared five volatile components (*β*-pinene, *cis*-linalool-oxide, terpinolene, linalool, 1-terpinen-4-ol), which gave the highest EAD responses from the three EOs, thus they might play an important role in the deterrent effect. Although the release rate of the five shared, antennal active compounds was not constant, the behavioural effect of the EOs did not change significantly during the bioassay. It is well known that every liquid vaporizes to a different extent until it reaches equilibrium with its vapour above. At equilibrium state, the same number of molecules vaporize into the headspace, as the number of molecules condensing from the headspace into the liquid [56]. However, EOs are highly complex mixtures, constituents with different vapor pressure have a competition to evaporate. Compounds with high vapor pressure are capable of suppressing the vapor pressure of fellow constituents in a closed environment. It is also well accepted that, once deprived of the protective compartmentation in the plant matrix [57], essential oil constituents especially tend to sustain oxidative damage, chemical transformations or polymerization reactions [58]. Thanks to the structural relationship within the same chemical group, essential oil constituents easily convert into each other by oxidation, isomerization, cyclization, dehydrogenation reactions [58]. For instance, terpenoids tend to be both volatile and thermolabile and may be easily oxidized or hydrolyzed depending on their respective structure [59].

The equilibrium in an open headspace can also be different from minute to minute and with changes in the composition of these mixtures, the matrix effects can also change. These effects and the interaction can easily account for the observed phenomenon, so that even the same compound can have different emission timespans in different EOs. Our results correspond with Molnár et al. [15], who also showed that the number of volatile components, emitted by the same type of dispenser, decreased during the measured period.

## 5. Conclusions

In conclusion, our results show that all three EOs have a significant oviposition-deterring effect. The antennal active compounds should be investigated in order to identify the one or ones that may be responsible for the deterrence and to understand whether the identified volatiles act additively or synergistically, or whether single compounds are already sufficient to deter oviposition. The efficiency of EOs or a synthetic mixture of the identified components should also be confirmed under field conditions. Ultimately, we hope our results may assist in understanding the chemo-ecological characteristics of this invasive moth species and pave the way to developing successful and environmentally sound control methods against BTM—e.g., repellent EOs together with pheromone in a successful push and pull strategy—and preserve the box tree populations in Europe.

## Figures and Tables

**Figure 1 insects-11-00465-f001:**
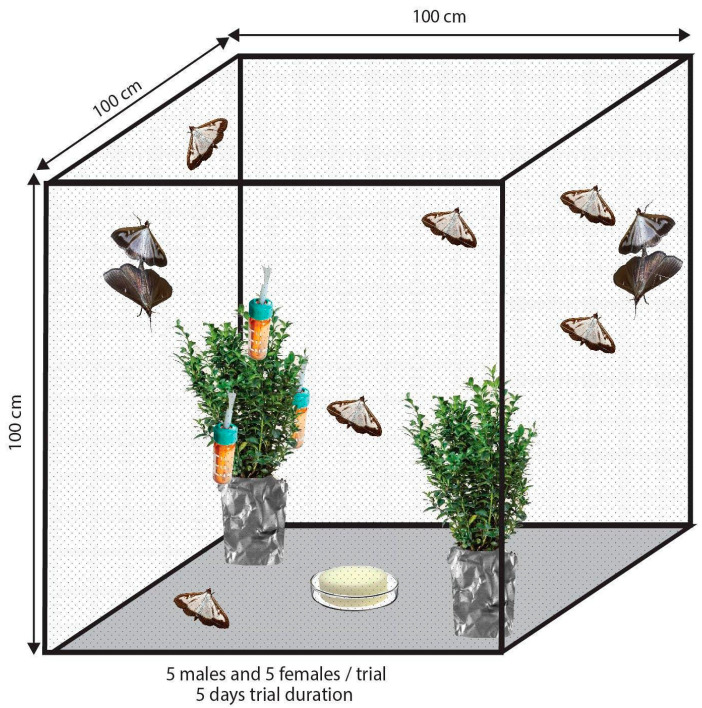
The scheme of the two-choice oviposition bioassay for adult box tree moths in screened cages with polyester netting. Two potted boxwood plants were offered for oviposition, one treated with essential oil-filled vial-wick dispensers compared against an untreated plant.

**Figure 2 insects-11-00465-f002:**
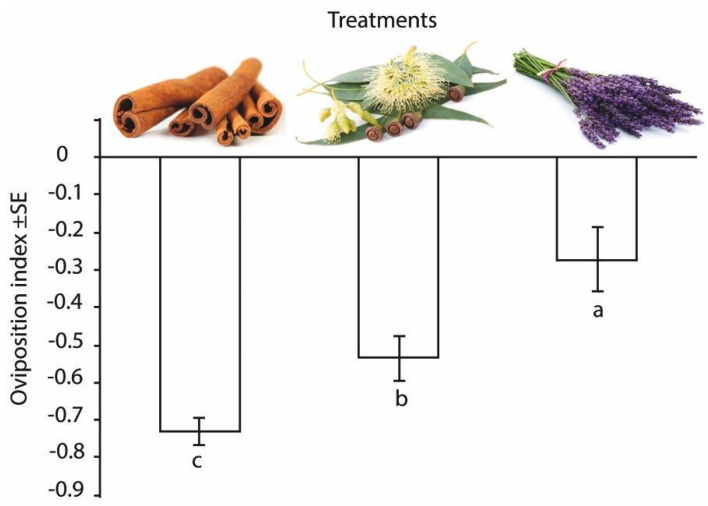
Oviposition indices (Oi) calculated from two-choice oviposition assays where boxwood plants treated with (**a**) lavender (**b**) eucalyptus and (**c**) cinnamon, and essential oil filled vial-wick dispensers were offered against untreated plants as oviposition sites for box tree moths (*Cydalima perspectalis*). Columns differing significantly (ANOVA, Games–Howell post hoc test, *p* < 0.05) are labelled.

**Figure 3 insects-11-00465-f003:**
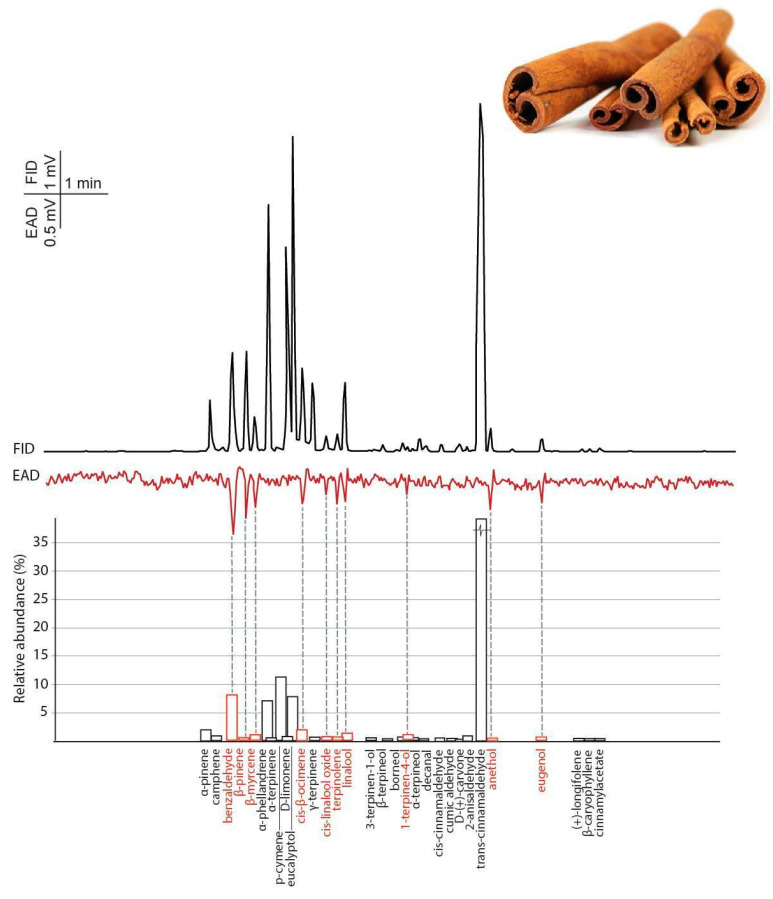
The averaged antennal response (EAD) of female box tree moths (*Cydalima perspectalis*) to cinnamon essential oil headspace samples (collected with SPME) measured by gas chromatography coupled electroantennographic detection (GC-EAD) (n = 5). Bars indicate the relative abundance of volatile constituents based on GC-MS analysis. The names of antennal active volatiles are highlighted in red.

**Figure 4 insects-11-00465-f004:**
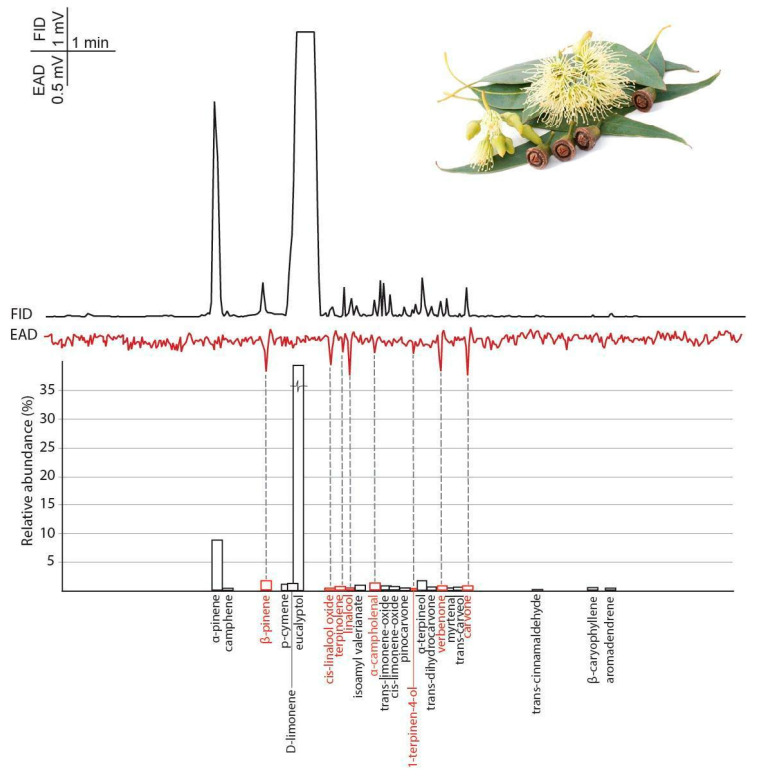
The averaged antennal response (EAD) of female box tree moths (*Cydalima perspectalis*) to eucalyptus essential oil headspace samples (collected with SPME) measured by gas-chromatography-coupled electroantennographic detection (GC-EAD) (n = 5). Bars indicate the relative abundance of volatile constituents based on GC-MS analysis. The names of antennal active volatiles are highlighted in red.

**Figure 5 insects-11-00465-f005:**
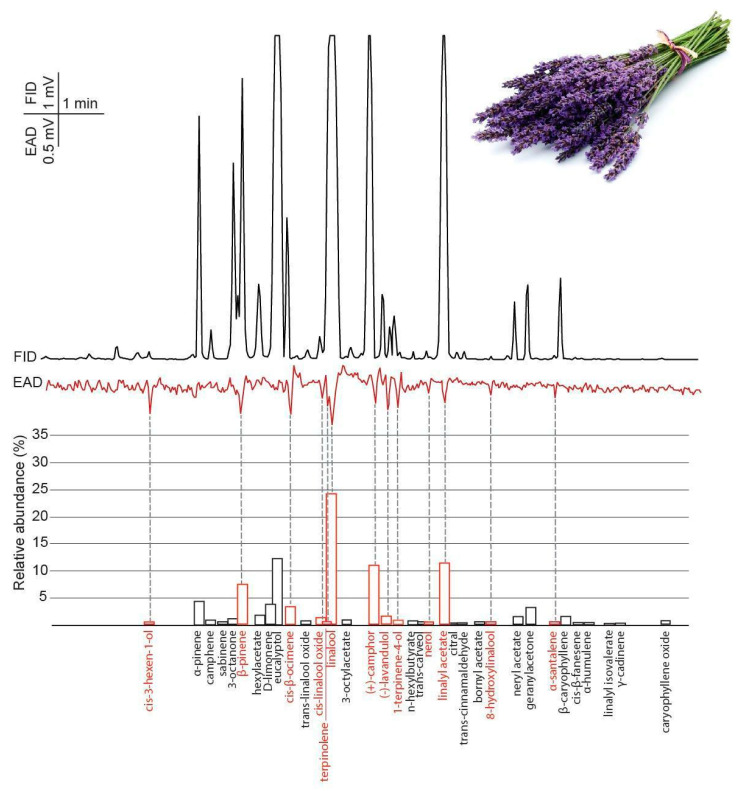
The averaged antennal response (EAD) of female box tree moths (*Cydalima perspectalis*) to lavender essential oil headspace samples (collected with SPME) measured by gas chromatography coupled electroantennographic detection (GC-EAD) (n = 5). Bars indicate the relative abundance of volatile constituents based on GC-MS analysis. The names of antennal active volatiles are highlighted in red.

**Table 1 insects-11-00465-t001:** Antennal active volatile compounds with their relative content in the three EOs’ headspace. RI NIST: retention index obtained from NIST database, RI Calc.: calculated RI, ΔRI: absolute difference of RI NIST and RI Calc.

						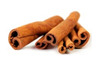	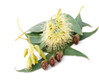	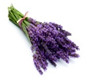
#	RI NIST	RI Calc.	∆RI	CAS	Compounds	Relative Content (%)
1	855	860	5	928-96-1	*cis*-3-hexen-1-ol	-	-	0.03
2	966	969	3	100-52-7	benzaldehyde	8.23	-	-
3	980	988	8	127-91-3	*β*-pinene	0.20	1.54	6.70
4	992	997	5	123-35-3	*β*-myrcene	0.46	-	-
5	1037	1056	19	3338-55-4	*cis-β*-ocimene	1.50	-	3.32
6	1078	1080	2	5989-33-3	*cis*-linalool oxide	0.05	0.11	0.672
7	1090	1096	6	586-62-9	terpinolene	0.18	0.57	0.34
8	1100	1104	4	78-70-6	linalool	0.80	0.05	24.07
9	1140	1162	18	91,819-58-8	*α*-campholenal	-	0.77	-
10	1150	1140	10	464-49-3	(+)-camphor	0.26	-	11.70
11	1165	1176	11	nd	(-)-lavandulol	-	-	0.06
12	1177	1192	15	562-74-3	1-terpinen-4-ol	0.98	0.06	0.39
13	1206	1223	17	80-57-9	verbenone	-	0.40	-
14	1239	1238	1	106-25-2	nerol	-	-	0.05
15	1254	1256	2	99-49-0	carvone	-	0.66	-
16	1257	1268	11	115-95-7	linalyl acetate	-	-	11.76
17	1289	1295	6	104-46-1	anethol	0.52	-	-
18	1355	1356	1	64,142-78-5	8-hydroxylinalool	-	-	0.01
19	1359	1367	8	97-53-0	eugenol	0.01	-	-
20	1420	1437	17	512-61-8	*α*-santalene	-	-	0.03

**Table 2 insects-11-00465-t002:** Results of univariate GLM testing release rates for 5 consecutive days.

		Coefficients	Std. Error	*p* Value	95% Confidence Interval	
Cinnamon	*β*-pinene	58,508	9358	0.000 *	38,293	78,724
	*cis*-linalool oxide	763,329	200,623	0.002 *	329,909	1,196,749
	terpinolene	2,867,102	1,109,690	0.023 *	469,763	5,264,441
	linalool	186,612	82,758	0.042 *	7825	365,399
	1-terpinen-4-ol	136,870	21,533	0.000 *	90,351	183,389
Eucalyptus	*β*-pinene	−6,317,955	3,663,618	0.108	−14,232,721	1,596,811
	*cis*-linalool oxide	−2,130,450	355,777	0.000 *	−299,058	−1,361,841
	terpinolene	−305,624	112,878	0.018 *	−549,481	−61,767
	linalool	−258,880	95,804	0.018 *	−465,851	−51,908
	1-terpinen-4-ol	−469,621	89,026	0.000 *	−661,949	−277,293
Lavender	*β*-pinene	−4,545,632	1,623,038	0.015 *	−8,051,994	−1,039,271
	*cis*-linalool oxide	−247,724	81,186	0.009 *	−423,115	−72,333
	terpinolene	−13,890,875	7,393,673	0.088	−29,863,934	2,082,183
	linalool	−5,950,558	1,698,063	0.004 *	−9,618,999	−2,282,117
	1-terpinen-4-ol	−347,161	153,093	0.041 *	−677,897	−16,424

* indicate significant cases.

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
