# Peer review of "Essential Oil Headspace Volatiles Prevent Invasive Box Tree Moth (Cydalima perspectalis) Oviposition—Insights from Electrophysiology and Behaviour"

_insects, 2020, doi:10.3390/insects11080465_

Round 1

Reviewer 1 Report

First, I would like to congratulate all authors for their theme selection and experimental documentation. Nevertheless, I would like to indicate some points were there are weaknesses, which my following comments are aiming to resolve. I feel that although those comments and suggestions are limited a lengthy period will be required therefore I suggest a major revision.

Comments

  1. Abstract (line 17): …are hardly available for this insect. Since you recognize availability site it, otherwise correct: …are not available…
  2. Introduction (line 53): … are known. For what are known for? Please amend.
  3. Introduction (line 56): … 20-60 components… Essential oils may contain less than 20 and above 60 compounds, belonging to various chemical classes e.g unsaturated H/C, aromatic compounds etc. Please consider rewriting this paragraph including more updated and relevant references.
  4. Introduction (lines 60-65): In this paragraph authors summarize the vast amount of research on the insect-essential oil interaction, and the still unresolved issue of efficacy through two 20th century references and a 2010 review. I strongly suggest rewriting this paragraph in order to indicate the progress in the relevant subjects, aiming also to highlight the relevant knowledge gap on Lepidoptera in general and BTM in specific.
  5. Materials and Methods (lines 117-135): Quantitative analysis of essential oil compounds based on MS is not credible and in order to provide sound quantitative results authors should either include their methodological approach for quantification established on MS or provide the relevant FID results. It is also stated here that Ki was calculated and was a crucial step for the essential oils’ components identification. If this is the case in Supplementary material’s Table authors should include the experimentally calculated Ki and not the ones from NIST. Furthermore, authors should amend the methodology in order to include the relevant agents and procedures for the Ki calculation.  

Author Response

Dear Reviewer,

We have revised our manuscript (insects-845723) following your suggestions and recommendations. We would like to thank the reviewer’s efforts. The suggestions and recommendations were very useful and helped us to refine the manuscript. We genuinely hope that you will find the revised version acceptable for publication in Insects.

We provided below a detailed list of our responses to all the comments and suggestions. For clarity purposes, after quoting each comment, our response is written in bold.

Thank you for your consideration,

Béla Péter Molnár

  1. Abstract (line 17): …are hardly available for this insect. Since you recognize the availability site it, otherwise correct: …are not available…

Line 19: Sentence is corrected according to the Reviewers suggestion.

  1. Introduction (line 53): … are known. For what are known for? Please amend.

The critical statement is deleted because the source publication did not clarify the method of identifying the number of essential oils, thank you for drawing our attention to the subject.

  1. Introduction (line 56): … 20-60 components… Essential oils may contain less than 20 and above 60 compounds, belonging to various chemical classes e.g unsaturated H/C, aromatic compounds etc. Please consider rewriting this paragraph including more updated and relevant references.

Line 56-64: The paragraph is rephrased according to the Reviewer suggestions, we highlighted better the underlying reasons for the chemical variability of EOs.

  1. Introduction (lines 60-65): In this paragraph authors summarize the vast amount of research on the insect-essential oil interaction, and the still unresolved issue of efficacy through two 20th century references and a 2010 review. I strongly suggest rewriting this paragraph in order to indicate the progress in the relevant subjects, aiming also to highlight the relevant knowledge gap on Lepidoptera in general and BTM in specific.

Line 64-76: The paragraph is rephrased according to the Reviewers suggestions; more recent publications are included to indicate the current research directions.

  1. Materials and Methods (lines 117-135): Quantitative analysis of essential oil compounds based on MS is not credible and in order to provide sound quantitative results authors should either include their methodological approach for quantification established on MS or provide the relevant FID results.

Thank you for pointing this out, but we would like to emphasize that SPME sampling is widely used as a qualitative extraction tool. Quantitative analyses are possible with SPME sampling but very complicated to calculate (for example: Larroque, V., Desauziers, V., & Mocho, P. (2006). Comparison of two solid-phase microextraction methods for the quantitative analysis of VOCs in indoor air. Analytical and bioanalytical chemistry, 386(5), 1457-1464, 2006). Although it might have avoided the Reviewer’s attention but qualitative analysis was carried out, quantitative analysis of the examined essential oils was not included in the text. Integrated areas under the compounds’ curves were used to compare compounds and perform statistical analyses.  For the GLM, we used the MS abundances, and with the analysis, we compared the abundance of the same compounds between different injections using the same settings in the GC and the same tuning file in the MS, hence this should not affect our results.

However, we added more details to the Material and Methods about the basis of the analysis (Line 160-161.).

  1. It is also stated here that Ki was calculated and was a crucial step for the essential oils’ components identification. If this is the case in Supplementary material’s Table authors should include the experimentally calculated Ki and not the ones from NIST. Furthermore, authors should amend the methodology in order to include the relevant agents and procedures for the Ki calculation.

Thank you for your remark, we added the calculated KI values to the Supplementary material (STable 1.) and rephrased the section (Line 145-149) of Materials and methods with the database used for comparison.

Reviewer 2 Report

Overall this is an interesting paper that shows results obtained in laboratory conditions with the use of essential oils as repellents toward Cydalima perspectalis, an invasive species in Europe that causes severe damage both in natural and ornamental boxwood. The paper is good in methodology and well written. However, there are a few things that should be better explained or clarified along the paper before it can be re-considered for publication. The most evident is that, from the results obtained, seems that the possible repellent effect determined by cinnamon and in a lesser extent by eucalyptus oils is likely determined by the minoritary compound present in these blends. The second is that, again from the results obtained, the assumption that the repellency effect of an essential oil is likely determined by the EAG active compound of the blend is not always true. These two aspects should be pointed out better in the discussion. Moreover some more details in the methodology of the experiments should be reported. English is good. The discussion can be improved and implemented with more references.

Line 44. Please give some more detail about why the use of pheromone for reducing population is not suitable.

Line 49-51, please rephrase this sentence, explaining better what happened in this research.

Line 52. This sentence should be linked better with the previous one.

Line 58-59. This is not always true, try to dampen this sentence.

Line 62, Cydia pomonella goes in italic and please add “L.”

Line 60-65. I think more case of the use of essential oils for insects of agricultural and stored products importance should be reported. See for example:

Lee, B. H., Choi, W. S., Lee, S. E., & Park, B. S. (2001). Fumigant toxicity of essential oils and their constituent compounds towards the rice weevil, Sitophilus oryzae (L.). Crop protection20(4), 317-320.

Ghabbari, M., Guarino, S., Caleca, V., Saiano, F., Sinacori, M., Baser, N., ... & Lo Verde, G. (2018). Behavior-modifying and insecticidal effects of plant extracts on adults of Ceratitis capitata (Wiedemann)(Diptera Tephritidae). Journal of Pest Science91(2), 907-917.

Benelli, G., Govindarajan, M., AlSalhi, M. S., Devanesan, S., & Maggi, F. (2018). High toxicity of camphene and γ-elemene from Wedelia prostrata essential oil against larvae of Spodoptera litura (Lepidoptera: Noctuidae). Environmental Science and Pollution Research25(11), 10383-10391.

Line 67. Report also the scientific names of the species used.

Line 987 Oviposition bioassay. What is the duration of the trial? Maybe I miss it. Moreover: what is the material of the cage? Net? Plexiglas with net holes? This is important to report because if the cage is a closed system, saturation can occur.

Line 199. The results don’t correspond to the graph. Trans-cinnamaldehyde amount look much higher than 5.25%. Please check.

Line 239. Temporal changes of the dispenser emitted volatiles. Personally I would have preferred to do the experiments differently: checking the emission from the releasers at a larger interval of times, eg 0-3-9-12-15 days. In fact I don’t think that the use of such releaser can be feasible if I have to change them every 5 days. Moreover also the table 2 that presents the results is not to me immediately clear. GLM analysis is correct, however, in my opinion a simple graph with histograms showing the mean emission during these five days would be clearer. I also think that is better do divide more clearly the compounds emission from each species. Finally, I found unusual that the emission of a same compound decreases in one essential oil and increase its emission in another essential oil, possible explanations of this should be clearly explained later on in the discussion.

Line 242, eucalyptol = Eucalyptus?

Line 261- Drosophila melanogaster goes in italic.

Line 267, I think that the authors should point out that 1,8-cineol is commonly known as eucalyptol, as before was named “eucalyptol”.

Line 288-290. In reality lavender was the less effective essential oil used even if in this one is present the highest amount of EAG active compounds. This should be pointed out. Moreover, see also other references that report EAG reponses to such compounds:

Zito, P., Guarino, S., Peri, E., Sajeva, M., & Colazza, S. (2013). Electrophysiological and behavioural responses of the housefly to “sweet” volatiles of the flowers of Caralluma europaea (Guss.) NE Br. Arthropod-Plant Interactions7(5), 485-489.

Meza, F. C., Roberts, J. M., Sobhy, I. S., Okumu, F. O., Tripet, F., & Bruce, T. J. (2020). Behavioural and Electrophysiological Responses of Female Anopheles gambiae Mosquitoes to Volatiles from a Mango Bait. Journal of chemical ecology, 1-10.

Overall I think the discussion should be improved. In the discussion should be pointed out that the qualitative differences between the EOs tested determined the different results in terms of oviposition deterrency. For example minoritary compounds present in cinnamon oils, still with EAG activity as myrcene or terpineol are not present in lavender, and was part of the blend of other essential oil that have ovicidal acitivity towart other insects, see for example Bedini, S., Guarino, S., Echeverria, M. C., Flamini, G., Ascrizzi, R., Loni, A., & Conti, B. (2020). Allium sativum, Rosmarinus officinalis, and Salvia officinalis essential oils: A spiced shield against blowflies. Insects11(3), 143. See also Yang, Y. C., Choi, H. Y., Choi, W. S., Clark, J. M., & Ahn, Y. J. (2004). Ovicidal and adulticidal activity of Eucalyptus globulus leaf oil terpenoids against Pediculus humanus capitis (Anoplura: Pediculidae). Journal of Agricultural and Food Chemistry52(9), 2507-2511.

Finally in the perspectives I would suggest the possible application of such repellents together with pheromones used in traps in roder to accomplish a push and pull strategy. See Cook, S. M., Khan, Z. R., & Pickett, J. A. (2007). The use of push-pull strategies in integrated pest management. Annu. Rev. Entomol.52, 375-400.

Author Response

Dear Reviewer,

We have revised our manuscript (insects-845723) following your suggestions and recommendations. We are very thankful for all the comments and the constructive critics that were aimed to improve our manuscript. We have carefully performed all the suggested changes and answered questions. We hope that you will find this revised version acceptable for publication in the Insects.

We provided below a detailed list of our responses to all the comments and suggestions. For clarity purposes, after quoting each comment, our response is written in bold and we use the line numbering of the document.

Thank you for your consideration,

Béla Péter Molnár

First of all, we would like to thank you and reply to the comment on the limitations of GC-EAD. It is an important aspect of GC-EAD, which should always be kept in mind. But we genuinely believe that the high throughput and fast pace of these measurements make up for the well-known limitations of antennographic detection and it is still a useful tool in understanding the olfactory background of deterrence.

  1. Line 44. Please give some more detail about why the use of pheromone for reducing population is not suitable.

We have rephrased that sentence and provided more details on this topic (Line 44-47).

  1. Line 49-51, please rephrase this sentence, explaining better what happened in this research.

We rephrased the sentence explaining better the results of the cited experiment (Line 52-55).

  1. Line 52. This sentence should be linked better with the previous one.

The whole paragraph is already rephrased based on the recommendation from Reviewer 1.

  1. Line 58-59. This is not always true, try to dampen this sentence.

The whole paragraph is already rephrased based on the recommendation from Reviewer 1. Thank you for pointing out the missing link between the paragraphs, and the problematic phrasing. Both sentences are rephrased according to the Reviewers suggestions.

  1. Line 62, Cydia pomonella goes in italic and please add “L.”

Thank you for your observation. However this example was removed from the text in the meantime.

Line 60-65. I think more case of the use of essential oils for insects of agricultural and stored products importance should be reported. See for example:

Lee, B. H., Choi, W. S., Lee, S. E., & Park, B. S. (2001). Fumigant toxicity of essential oils and their constituent compounds towards the rice weevil, Sitophilus oryzae (L.). Crop protection, 20(4), 317-320.

Ghabbari, M., Guarino, S., Caleca, V., Saiano, F., Sinacori, M., Baser, N., ... & Lo Verde, G. (2018). Behavior-modifying and insecticidal effects of plant extracts on adults of Ceratitis capitata (Wiedemann)(Diptera Tephritidae). Journal of Pest Science, 91(2), 907-917.

Benelli, G., Govindarajan, M., AlSalhi, M. S., Devanesan, S., & Maggi, F. (2018). High toxicity of camphene and γ-elemene from Wedelia prostrata essential oil against larvae of Spodoptera litura (Lepidoptera: Noctuidae). Environmental Science and Pollution Research, 25(11), 10383-10391.

The paragraph is rephrased according to the Reviewers suggestions; more recent publications are included to indicate the current research directions, and we also included more examples for using EOs in pest management. (Lines 64-76)

  1. Line 67. Report also the scientific names of the species used.

Although they were reported in the Materials and Methods, we moved the scientific names into Line 67, it is certainly more logical, thank you.

  1. Line 987 Oviposition bioassay. What is the duration of the trial? Maybe I miss it.

The experiment lasted for 5 consecutive days as it is explained in the line 123.

  1. Moreover: what is the material of the cage? Net? Plexiglas with net holes? This is important to report because if the cage is a closed system, saturation can occur.

We have not clarified the materials of the cages, indeed, thank you for your reflection, it is now corrected in line 111:

... in two screened cages with polyester netting (100 × 100 × 100 cm)

  1. Line 199. The results don’t correspond to the graph. Trans-cinnamaldehyde amount look much higher than 5.25%. Please check.

Thank you for pointing out the inaccurate relative abundances of the substituents written in the text. Relative abundances are all corrected according to the figures.

  1. Line 239. Temporal changes of the dispenser emitted volatiles. Personally I would have preferred to do the experiments differently: checking the emission from the releasers at a larger interval of times, eg 0-3-9-12-15 days. In fact I don’t think that the use of such a release can be feasible if I have to change them every 5 days.

The Reviewer is right, no doubt, if we want to evaluate this type of dispenser for long term usage, we definitely have to check the emission at least for 2 weeks. However, the underlying reason for measuring the temporal changes was to monitor if there were changes in the abundance of the emitted volatiles during the oviposition bioassay.

  1. Moreover also the table 2 that presents the results is not to me immediately clear. GLM analysis is correct, however, in my opinion a simple graph with histograms showing the mean emission during these five days would be clearer. I also think that is better do divide more clearly the compounds emission from each species.

We believe that the table with the GLM results gives more information about the trends in the changes of different compounds, however, we included the requested graph in the Supplement Materials to show the trends visually too.

  1. Finally, I found unusual that the emission of a same compound decreases in one essential oil and increase its emission in another essential oil, possible explanations of this should be clearly explained later on in the discussion.

Thank you for pointing out the phenomenon that compounds can behave differently in mixtures with different composition. We believe that temporal changes in the abundances of certain compounds of an essential oil headspace are influenced by many factors: chemical structure, chemical structure of the fellow constituents, quantity of fellow compounds, vapour pressure of other constituents.

Every liquid vaporizes to a different extent until it reaches equilibrium with its vapour above. At equilibrium state, the same amount of molecules vaporize into the headspace, as the number of molecules condensing from the headspace into the liquid. The faster these phase transitions occur in case of a compound, the greater its vapour pressure is at a given temperature (Eric Stauffer, Julia A. Dolan, Reta Newman 2008: chapter 4.6.5. Vapor Pressure pp. 105-127. In: Fire Debris Analysis. Elsevier). It is well accepted that once deprived of the protective compartmentation in the plant matrix (Treibs 1956 - Treibs W. 1956. Gewinnung atherischer ¨ Ole durch Destillation, Extraktion ¨ und Pressung. A. Einleitung. In: Treibs W, editor. E. Gildemeister / Fr. Hoffmann: Die atherischen ¨ Ole Band I. 4th ed. Berlin, Germany: ¨ Akademie-Verlag. p 307–9.) essential oil constituents especially tend to sustain oxidative damage, chemical transformations or polymerization reactions (Turek and Stintzing, 2013 -  Claudia Turek and Florian C. Stintzing (2013) Stability of Essential Oils: A Review Comprehensive Reviews in Food Science and Food Safety Free Access. Vol. 12. https://doi.org/10.1111/1541-4337.12006). Thanks to the structural relationship within the same chemical group, essential oil constituents are known to easily convert into each other by oxidation, isomerization, cyclization, dehydrogenation reactions (Turek and Stintzing, 2013). For instance terpenoids tend to be both volatile and thermolabile and may be easily oxidized or hydrolyzed depending on their respective structure (Scott 2005 - Scott RPW. 2005. Essential oils. In: Worsfold P, Townshend A, Poole C, editors. Encyclopedia of analytical science. 2nd ed. Amsterdam, London, New York: Elsevier. p 554–61.).

The equilibrium in an open headspace can also be different from minute to minute and with changes in the composition of these mixtures, the matrix effects can also change. These effects and the interaction can easily account for the observed phenomenon, that even the same compound can have different emission timespan in different EOs.

  1. Line 242, eucalyptol = Eucalyptus?

Eucalyptol is corrected to eucalyptus.

  1. Line 261- Drosophila melanogaster goes in italic.

It is reformatted, and also the name of the describer is added (Line 312).

  1. Line 267, I think that the authors should point out that 1,8-cineol is commonly known as eucalyptol, as before was named “eucalyptol”.

1,8-cineol is substituted with eucalyptol in Line 317.

  1. Line 288-290. In reality lavender was the less effective essential oil used even if in this one is present the highest amount of EAG active compounds. This should be pointed out. Moreover, see also other references that report EAG reponses to such compounds:

Zito, P., Guarino, S., Peri, E., Sajeva, M., & Colazza, S. (2013). Electrophysiological and behavioural responses of the housefly to “sweet” volatiles of the flowers of Caralluma europaea (Guss.) NE Br. Arthropod-Plant Interactions, 7(5), 485-489.

Meza, F. C., Roberts, J. M., Sobhy, I. S., Okumu, F. O., Tripet, F., & Bruce, T. J. (2020). Behavioural and Electrophysiological Responses of Female Anopheles gambiae Mosquitoes to Volatiles from a Mango Bait. Journal of chemical ecology, 1-10.

Thank you for helping to improve the discussion. We have included a more detailed discussion of the lavender oil and its lower efficiency and the probable underlying causes. 

  1. Overall I think the discussion should be improved. In the discussion should be pointed out that the qualitative differences between the EOs tested determined the different results in terms of oviposition deterrency. For example minoritary compounds present in cinnamon oils, still with EAG activity as myrcene or terpineol are not present in lavender, and was part of the blend of other essential oil that have ovicidal acitivity towart other insects, see for example Bedini, S., Guarino, S., Echeverria, M. C., Flamini, G., Ascrizzi, R., Loni, A., & Conti, B. (2020). Allium sativum, Rosmarinus officinalis, and Salvia officinalis essential oils: A spiced shield against blowflies. Insects, 11(3), 143. See also Yang, Y. C., Choi, H. Y., Choi, W. S., Clark, J. M., & Ahn, Y. J. (2004). Ovicidal and adulticidal activity of Eucalyptus globulus leaf oil terpenoids against Pediculus humanus capitis (Anoplura: Pediculidae). Journal of Agricultural and Food Chemistry, 52(9), 2507-2511

Thank you for pointing out this aspect of our results obtained on eucalyptus and cinnamon oil. As we did not analyze dose-dependency of the deterrence elicited and we did not analyze the behavioural effect of each EAD-active component separately we cannot conclude that only the minor compounds were responsible for the altered behaviour. It is also important that the behavioural effect of constituents in a volatile mixture is not always additive as congruent and incongruent information can synergise or antagonise each other (REF). We also tried to make this clear in the discussion.

  1. Finally in the perspectives I would suggest the possible application of such repellents together with pheromones used in traps in order to accomplish a push and pull strategy.

See Cook, S. M., Khan, Z. R., & Pickett, J. A. (2007). The use of push-pull strategies in integrated pest management. Annu. Rev. Entomol., 52, 375-400.

Thank you for this suggestion, we implemented into the conclusions.

Round 2

Reviewer 1 Report

Honorable colleagues I do no perceive the corrections you indicate as sufficient resolution of my concerns. 

Author Response

Dear Reviewer,

We have tried again to correct our manuscript but the comments were not specified.

We provided below a detailed list of our responses to all the comments and suggestions. For clarity purposes, after quoting each comment, our response is written in bold, and we also copied the original paragraphs and the new or rephrased paragraphs.

We genuinely hope that you will find the revised version acceptable for publication in Insects.

Thank you for your consideration,

Béla Péter Molnár

  1. Abstract (line 17): …are hardly available for this insect. Since you recognize the availability site it, otherwise correct: …are not available…

Sentence is corrected according to the Reviewers suggestion (Line 18) because we could not add a solid reference here.

Abstract: The box tree moth (Cydalima perspectalis Walker) is an invasive species in Europe causing severe damage both in natural and ornamental boxwood (Buxus spp.) vegetation. Pest management tactics are often based on the use of chemical insecticides, whereas environmentally-friendly control solutions are not available against this insect.”

  1. Introduction (line 53): … are known. For what are known for? Please amend.

The critical statement is deleted because the source publication did not clarify the method of identifying the number of essential oils, thank you for drawing our attention to the subject.

The original paragraph is fundamentally rephrased:

“Thousands of plants have been screened as potential sources of repellents and/or insecticides in the last 50 years (Sukumar et al. 1991), and approximately 3000 essential oils (EOs) are known. Essential oils are plant-derived, concentrated hydrophobic liquids containing volatile chemical compounds. These complex volatile mixtures can be isolated from a vast array of plants, and they contain as many as 20-60 components belonging to hydrocarbons (terpenes and sesquiterpenes) and oxygenated compounds (alcohols, esters, ethers, aldehydes, ketones, lactones, phenols, and phenol ethers) (Guenter, 1972; Bakkali et al., 2008). Besides their complexity, EOs can be characterized by two or three major components of relatively high concentration (20-70%) (Bakkali et al., 2007). “

New paragraph (Lines 56-65)

“For potential sources of repellents, thousands of plants have been screened in the past 50 years [19]; and recently essential oils became the focus of attention. These concentrated hydrophobic liquids can be isolated from a vast array of plants, and the composition highly depends on the genotype [20], on the phenological stage [21], on the harvested plant part [22] and on the extraction method [23]. The volatile components are biosynthesized in different pathways, so they belong to polyketides and lipids, shikimic acid derivatives and terpenoids (hemi-, mono, sesquiterpenoids) [15,16,20]. These complex mixtures may contain over 300 different compounds [24], generally with low molecular weight [25]. In spite of their complexity, most of the EOs are dominated by two or three major components while the rest of the compounds are present only in trace amounts [20].”

  1. Introduction (line 56): … 20-60 components… Essential oils may contain less than 20 and above 60 compounds, belonging to various chemical classes e.g unsaturated H/C, aromatic compounds etc. Please consider rewriting this paragraph including more updated and relevant references.

The paragraph is rephrased according to the Reviewer suggestions, we highlighted better the underlying reasons for the chemical variability of EOs. See also answer to question number 2.

The original paragraph is rephrased:

“These complex volatile mixtures can be isolated from a vast array of plants, and they contain as many as 20-60 components belonging to hydrocarbons (terpenes and sesquiterpenes) and oxygenated compounds (alcohols, esters, ethers, aldehydes, ketones, lactones, phenols, and phenol ethers) (Guenter, 1972; Bakkali et al., 2008). Besides their complexity, EOs can be characterized by two or three major components of relatively high concentration (20-70%) (Bakkali et al., 2007). “

New paragraph (Lines 57-61)

“These concentrated hydrophobic liquids can be isolated from a vast array of plants, and the composition highly depends on the genotype [20], on the phenological stage [21], on the harvested plant part [22] and on the extraction method [23]. The volatile components are biosynthesized in different pathways, so they belong to polyketides and lipids, shikimic acid derivatives and terpenoids (hemi-, mono, sesquiterpenoids) [15,16,20]. These complex mixtures may contain over 300 different compounds [24], generally with low molecular weight [25]. In spite of their complexity, most of the EOs are dominated by two or three major components while the rest of the compounds are present only in trace amounts [20].”

  1. Introduction (lines 60-65): In this paragraph authors summarize the vast amount of research on the insect-essential oil interaction, and the still unresolved issue of efficacy through two 20th century references and a 2010 review. I strongly suggest rewriting this paragraph in order to indicate the progress in the relevant subjects, aiming also to highlight the relevant knowledge gap on Lepidoptera in general and BTM in specific.

Line 69-87: The paragraph is rephrased according to the Reviewers suggestions; more recent publications are included to indicate the current research directions. We also tried to highlight the difficulty of using EOs as pesticides and the possible solutions.

The original paragraph is rephrased:

“Essential oils were first proven to be effective against several haematophagous insects [26]. Their application, however, is not limited to these insect taxa. Essential oils are widely used as fumigants e.g. towards the rice weevil Sitophilus oryzae L. [27], antifeedants e.g against the lepidopteran Trichoplusia ni Hübn. [28], and oviposition deterrents e.g to keep the medfly (Ceratitis capitata Wiedemann) away [29]. Researches on oviposition deterrence against lepidopteran species such as Phthorimaea operculella Zell. [30], Anticarsia gemmatalis Hübn. [31], Spodoptera littoralis Boisd. [32], and  Spodoptera frugiperda J.E. Smith [33] showed that EOs have the potential to modify the egg-laying behaviour. To our knowledge, only one study was dedicated to investigating the effect of EOs to the ovipositing behaviour of BTM [12]. Some plant-based repellents are comparable to, or even more effective than synthetic volatile blends, but EOs tend to be short-lived in their efficacy. Consequently, development of dispensers for long-lasting EO emission becomes more and more important both in plant protection [34] and food industry [35].”

New paragraph (Lines)

“Essential oils were first proven to be effective against several haematophagous insects such as malaria mosquitoes (Anopheles sp. Meigen) [26], tsetse flies (Glossina sp. Wiedemann) [26] and ticks (Ixodes ricinus L.) [27]. Their application, however, is not limited to these insect taxa. Antifeedant effect of EOs was also studied against lepidopteran (Trichoplusia ni Hübn. [28]) and coleopteran species (Colorado potato beetle) [29], however, contact application on plants is limited due to the phytotoxic properties of EOs. In fumigation, the volatile compounds are responsible for lethality. Fumigant efficiency of different EOs was investigated against storage pests e.g. towards the rice weevil Sitophilus oryzae L. [30]  and Plutella xylostella L. [31]. To investigate the contact application of EOs regarding their oviposition deterrence, researches were concluded e.g on medfly (Ceratitis capitata Wiedemann) [32] and on cowpea beetle (Callosobruchus maculatus F.) [33]. Studies were also conducted on lepidopteran species such as Phthorimaea operculella Zell. [34], Anticarsia gemmatalis Hübn. [35], Spodoptera littoralis Boisd. [36], Spodoptera frugiperda J.E. Smith [37], and Tuta absoluta Meyrick [38] showing that EOs have the potential to modify the egg-laying behaviour both as contact and as volatile deterrents. Concerning the ovipositing behaviour of BTM, only one study investigated the contact ovipositing deterrent effect of different EOs [12].

Some plant-based repellents are comparable to, or even more effective than synthetic volatile blends, but EOs can be phytotoxic, and in dispensers, they tend to be short-lived in their efficacy. Consequently, development of dispensers for long-lasting EO emission both indoor and outdoor becomes more and more important both in the food industry [39] and plant protection, like encapsulated lemongrass EO [40], or nanoemulsion of eucalyptus oil [41].”

  1. Materials and Methods (lines 117-135): Quantitative analysis of essential oil compounds based on MS is not credible and in order to provide sound quantitative results authors should either include their methodological approach for quantification established on MS or provide the relevant FID results.

Thank you for pointing this out, but we would like to emphasize that we did not perform quantitative analysis during our experiments. Our goal was to calculate the relative abundances, and for this purpose, using total ion chromatogram is accepted. We did not compare the abundance of different compounds based on their total ion peak integral, we only compared the abundance of the same compound to itself in other lures. As the fragmentation pattern should be the same, the peak of the compound can be compared between EOs.

Although FID could be used for quantification, with SPME is extremely complicated, especially for complex mixtures like headspace of EOs (Larroque et al. 2006).

However, we added more details to the Material and Methods about the basis of the analysis (Line 160-161.).

The original paragraph is rephrased:

“For performing statistical analysis and calculating relative abundances, integrated areas of the compounds were used. Manual integration was adapted to the TIC baseline. “

New paragraph (Lines)

“For performing statistical analysis and calculating relative abundances, manually integrated areas from the total ion chromatogram adapted to the baseline were used. “

  1. It is also stated here that Ki was calculated and was a crucial step for the essential oils’ components identification. If this is the case in Supplementary material’s Table authors should include the experimentally calculated Ki and not the ones from NIST. Furthermore, authors should amend the methodology in order to include the relevant agents and procedures for the Ki calculation.

Thank you for your remark, beside the provided Kováts indices, obtained from NIST Library, we added the calculated Kováts indices - C6-C17 alkane mixture (see the table below) -  as well as relative differences between obtained and calculated RI for Table 1. and for Supplementary material (STable 1.). Subsequently the corresponding section of Material and Methods was also rephrased (Lines 154-161).

Retention times of alkane standards used to calculate Kováts indices of the tested essential oil constituents

n-alkanes

(number of carbon)

Rt

 (Retention time, min)

6

2.00

7

2.58

8

3.57

9

4.93

10

6.46

11

8.00

12

9.52

13

10.92

14

12.24

15

13.50

16

14.71

17

15.86

Table 1. Antennal active volatile compounds with their relative content in the three EOs’ headspace. RI NIST: retention index obtained from  NIST database, RI Calc.: calculated RI, ΔRI: absolute difference of RI NIST and RI Calc.

Reviewer 2 Report

the paper has been strongly improved, and is not suitable for pubblication 

very minor issue occurring in discussion:

Line 274, change Frabricius with F.

Line 287, please check

Line 303, delete ]

Line 316, Anopheles gambiae add descriptor

Author Response

Dear Reviewer,

We have corrected the minor issues in the discussion you have recommended. We are truly thankful for your patience, and for correcting our manuscript so thoroughly! We hope that after the corrections, you will feel our work is worth publishing.

Thank you for your consideration,

Béla Péter Molnár

  1. Line 274, change Frabricius with F.

Thank you for your suggestion, it is corrected.

  1. Line 287, please check

Thank you for drawing our attention to this! ßs are all replaced with βs and cis/trans are reformatted to italic!

  1. Line 303, delete ]

Thank you, ] is deleted.

  1. Line 316, Anopheles gambiae add descriptor
  2. Thank you, Giles is added after A. gambiae.

Round 3

Reviewer 1 Report

Dear authors, 

At the present time I am in a remote place collecting plants with limited internet access and therefore I am not able to perform the review...

Author Response

Dear Reviewer,

We would like to thank you for taking your time to read and correct our manuscript. We believe that all of your instructions, suggestions followed carefully and based on our best knowledge. Therefore, we genuinely feel, the manuscript has improved remarkably. 

We wish you productive fieldwork!

Sincerely,

Béla Péter Molnár
